# Modified Bose-Einstein condensation in an optical quantum gas

Mario Vretenar [1], Chris Toebes[1] & Jan Klaers [1✉]

Open quantum systems can be systematically controlled by making changes to their environment. A well-known example is the spontaneous radiative decay of an electronically excited emitter, such as an atom or a molecule, which is significantly influenced by the feedback from the emitter's environment, for example, by the presence of reflecting surfaces. A prerequisite for a deliberate control of an open quantum system is to reveal the physical mechanisms that determine its state. Here, we investigate the Bose-Einstein condensation of a photonic Bose gas in an environment with controlled dissipation and feedback. Our measurements offer a highly systematic picture of Bose-Einstein condensation under nonequilibrium conditions. We show that by adjusting their frequency Bose-Einstein condensates naturally try to avoid particle loss and destructive interference in their environment. In this way our experiments reveal physical mechanisms involved in the formation of a Bose-Einstein condensate, which typically remain hidden when the system is close to thermal equilibrium.

[1] Adaptive Quantum Optics (AQO), MESA+ Institute for Nanotechnology, University of Twente, PO Box 217, 7500 AE Enschede, The Netherlands.
✉email: j.klaers@utwente.nl

When a Bose gas exceeds the critical phase space density, a condensation process is triggered[1]. This process can be considered as a competition between various possible condensate wavefunctions that experience different growth rates under the given experimental conditions and in which ultimately the state with the highest rate prevails. If these growth rates are given by the respective Boltzmann factors, the system relaxes to thermal equilibrium and the condensation takes place in the state that minimizes the energy. It is possible to modify these rates by moving to non-equilibrium conditions, e.g., by creating a net particle flow between the condensate and its environment. Such a deliberate control of the growth rates in the condensation process can serve practical purposes, e.g., for the quantum simulation of spin models with optical condensates[2,3] and related systems[4,5], which is currently an active field of research.

If the conditions for reaching thermal equilibrium are not met, the condensation will not necessarily occur in the lowest energy state. Beyond that, the question arises: is it possible to formulate what kind of state a Bose condensate strives for under such non-equilibrium conditions? In contrast to cold atomic gases[6,7], which typically do not allow significant particle exchange with their environment, optical quantum gases in photonic[8–15] and polaritonic[16–18] microcavity systems offer the possibility of answering such questions experimentally. Previous studies of non-equilibrium condensation processes in optical microcavity systems have described, e.g., the influence of the pump geometry on the photon condensation and thermalization process[19,20], the occurrence of multimode condensation phenomena arising from a dynamical equilibrium between gain and loss[12,21–24], the non-equilibrium excitation spectrum of polaritonic condensates[25–27], and effects of bistability[28,29]. See refs. [30–32] for a broader overview.

In two-dimensional (2D) photonic quantum gases, photons obtain a non-zero chemical potential $\mu$, which, in thermal equilibrium, is related to the excitation level in the optical medium by $e^{\frac{\mu}{kT}} \propto \rho_\uparrow / \rho_\downarrow$[33]. Here, $\rho_{\uparrow,\downarrow}$ denote the densities of electronically excited and ground-state molecules, which can be set by optically pumping the medium. A complete thermalization process in the system, which can be created by repeated absorption and emission of photons by the optical medium[19,33], would make all net energy and particle flows disappear. In real systems with imperfect photon confinement, the thermalization process is not always able to bring the system into a global thermal equilibrium. This is especially true when highly inhomogeneous optical pump geometries are used. Such pump geometries can lead to non-equilibrium condensation phenomena in which the local chemical potential of the photons (understood as the local excitation level of the optical medium) varies across the system. This situation, however, still has to be differentiated from laser-like operation in which particle losses are comparable or even overcome photon reabsorption by the optical medium. Standard laser design typically seeks to prevent photon reabsorption, as this increases the lasing threshold. This can be achieved, e.g., by using a four-level laser scheme as opposed to a three-level laser scheme[34]. In the case of photon condensation experiments, on the other hand, one tries to prevent light from escaping from the resonator as much as possible (similar to a black body radiator). For this purpose, mirrors with very high reflectivity and an optical medium with sufficient spectral overlap between absorption and emission are used. The experiments described in this work are carried out in a parameter range in which the probability that a cavity photon is absorbed by the optical medium is significantly higher than the probability for transmission through the cavity mirrors, but not so high that the system relaxes into a

global state of equilibrium with spatially homogeneous chemical potential irrespective of the optical pumping geometry.

A particular class of non-equilibrium condensation phenomena are transport processes, which are a current research topic in 2D optical quantum gases[35–39]. In these systems, condensates propagating in the transverse plane of the resonator can either be created by directly condensing into states with higher kinetic energy, or by preparing condensates at rest and subsequently setting the particles in motion. To achieve the latter, the condensates must be generated in suitable potential landscapes using spatially inhomogeneous optical pump geometries. If the condensate is created at an elevated potential energy level, e.g., particles will gain kinetic energy as they fall into regions of lower potential. This can be used to prepare a stream of photons in a controlled state of motion. In our experiments, such photon currents are directed into a potential landscape that acts as a Mach–Zehnder interferometer. By partially or completely closing the outputs of the interferometer, we can systematically vary the degree of dissipation and feedback in the system, which allows us to identify the underlying physical principles that determine the formation of Bose–Einstein condensates under non-equilibrium conditions.

## Results and discussion

In our experiment, we use a high-finesse optical microcavity filled with a water-based solution of rhodamine 6G dye and the thermo-responsive polymer poly(N-isopropylacrylamide) (pNIPAM), in which the photons are repeatedly absorbed and re-emitted by the dye molecules (see Fig. 1a). This process is governed by the Kennard–Stepanov law linking the (broadband) absorption coefficient $B_{12}(\omega)$ and the emission coefficient $B_{21}(\omega)$ by $B_{12}(\omega)/B_{21}(\omega) = \exp[(\hbar(\omega - \omega_{zpl})/kT]$, where $\omega_{zpl}$ corresponds to the zero-phonon line of the dye and $T$ denotes the temperature. Low cavity losses allow the photons to be absorbed and re-emitted multiple times before leaving the cavity, creating a thermalization process between the photon gas and the dye solution. Indeed, Bose–Einstein distributed energies and the formation of Bose–Einstein condensates at room temperature have been demonstrated several times in this system[8,12,13,19,40]. The separation between the microcavity mirrors of $D_0 \simeq 10\,\mu m$ is sufficiently small so that the longitudinal mode number becomes a conserved quantity in the interaction between the photons and the optical medium[19]. In this regime, the photon gas effectively becomes 2D and the photon energy can be approximated by

$$E \simeq \frac{mc^2}{n_0^2} + \frac{(\hbar k_r)^2}{2m} - \frac{mc^2}{n_0^2}\left(\frac{\Delta d}{D_0} + \frac{\Delta n}{n_0}\right), \tag{1}$$

where $k_r$ is the transversal wavenumber describing the 2D motion in the cavity plane and $m$ denotes the effective photon mass. The second and third term correspond to the kinetic and potential energy. The potential energy is non-vanishing, if the distance between the mirrors $D(x, y) = D_0 + \Delta d(x, y)$ or the refractive index $n(x, y) = n_0 + \Delta n(x, y)$ is modified across the transverse plane of the resonator (we assume $\Delta d \ll D_0$ and $\Delta n \ll n_0$).

Three different experimental techniques are combined to control the potential landscape within the microresonator. First, a static nanostructuring of the mirror surfaces of the cavity mirrors is performed via a direct laser writing technique[41]. In this way, we can create smooth surface structures of up to 100 nm height with sub-nm precision in the growth direction. Second, a reversible tuning of the potential landscape is performed by locally heating the thermo-responsive polymer pNIPAM poly(N-isopropylacrylamide) that is added to the optical medium with a short laser pulse[40]. The absorbed laser energy increases the local temperature of the optical medium by a few Kelvin such that it

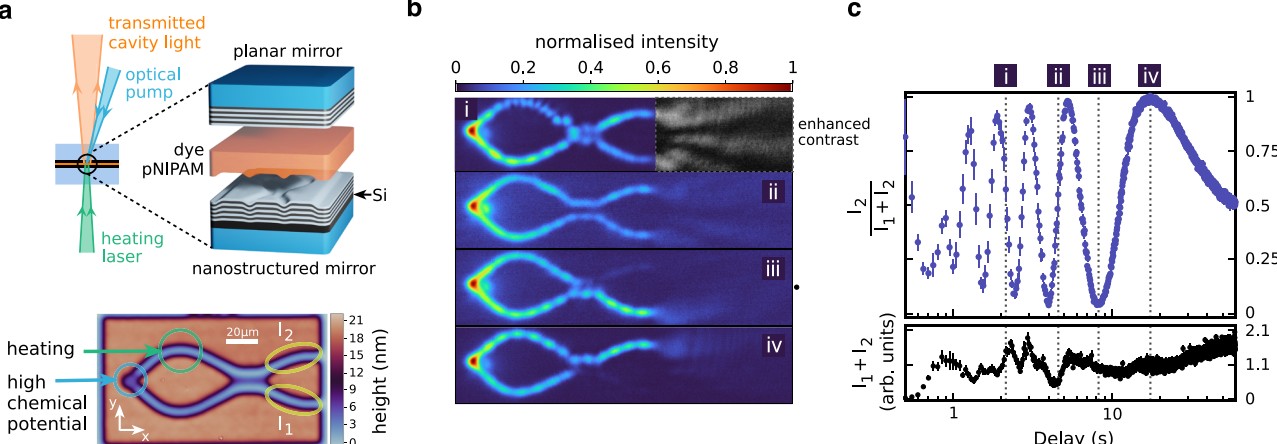

**Fig. 1 Photon Bose–Einstein condensation in an open Mach–Zehnder interferometer. a** Microcavity formed by a planar mirror, a nanostructured mirror, and an organic optical medium (top). The optical medium consists of a water-based solution of a dye (rhodamine 6G) and a thermo-responsive polymer (pNIPAM). The height profile of the nanostructured mirror (bottom) creates a potential landscape for the photon gas that effectively acts as a Mach–Zehnder interferometer. A laser beam at a wavelength of 532 nm is used to locally heat the thermo-responsive polymer. This allows for reversible tuning of the local refractive index, which can be used to create optical path length differences in the internal arms of the interferometer. The system is off-resonantly pumped with a tightly focused pulsed laser beam (pulse duration $\simeq 5$ ns, wavelength 470 nm). The time delay between the heating pulse and the optical pump pulse determines the path length difference that is probed. The photon density within the cavity is determined by measuring the transmitted cavity light with a camera. **b** Photon density for four time delays. **c** Normalized switching function $I_2/(I_1 + I_2)$ (upper graph) and total intensity $I_1 + I_2$ (lower graph) of the open interferometer as a function of the delay between heating and optical pumping (note the logarithmic time axis). Data points are averages over 20 measurements (optical pulses). Error bars indicate the SEM.

reaches the lower critical solution temperature (LCST) of pNI-PAM in water (32 °C). This leads to a significant increase of the index of refraction that, with Eq. (1), translates into a tunable potential for the photons in the microcavity. Finally, the potential landscape can be further adjusted by tilting one of the mirrors, effectively adding a potential gradient to the photon gas.

In our experiments, we use non-resonant optical pumping to create regions of high chemical potential in which photon Bose–Einstein condensates form (see location 'high chemical potential' in Fig. 1a). Particles emitted from the condensate into the microresonator plane are guided by two waveguide potentials. The waveguides are at a lower potential energy, which means that the condensate photons gain kinetic energy as they enter these regions (the mirror height at the condensate position is roughly 4.5 nm higher than the height level within the waveguide potential, see Fig. 1a). It is noteworthy that the photon Bose-Einstein condensate system differs here from many polaritonic systems: the potential gradient that leads to the acceleration of the particles is explicitly added by nanostructuring of the mirror. A particle–particle or particle–reservoir interaction that leads to a comparable effect in polaritonic systems[42,43] is negligible in the photon Bose-Einstein condensate system. The two waveguides first guide the photons into disjoint areas of the microresonator plane before photons recombine in a region that is designed as a 50:50 beamsplitter. In the following, the waveguides between the condensate and the beamsplitter structure are referred to as the internal arms of the interferometer, whereas the waveguides connected to the opposite end of the beamsplitter are referred to as output arms. Optical path length differences between the internal arms of the interferometer can be generated with the help of the thermo-responsive optical medium. For this purpose, we heat the upper internal arm with a focused laser pulse, see location 'heating' in Fig. 1a, so that the refractive index of the medium and thus the optical path length changes significantly. After the initial heating, the refractive index relaxes back to equilibrium. We observe that this decay is approximately exponential in time. Interestingly, the decay time is found to depend on the energy of

the initial heating pulse. For sufficiently strong heating, it can become as large as a few seconds.

Results for self-interference experiments with photon Bose-Einstein condensates performed in this manner are shown in Fig. 1b. Here we show the photon density in the microresonator plane for different time delays between the initial heating pulse and the optical pumping. Depending on the probed path length difference, a switching between constructive and destructive interference in the outputs can be observed. This observation confirms the coherence of the particle stream emitted by the photon Bose–Einstein condensate[44]. Another indication is the interference pattern that arises when the waveguide potentials at the end of the two outputs cease, which is reminiscent of that of a double slit experiment. A quantitative characterization of the system is obtained by determining the intensities $I_{1,2}$ in the lower and upper output arms of the interferometer (see locations '$I_1$' and '$I_2$' in Fig. 1a). The upper graph in Fig. 1c shows the normalized intensity in the upper output arm $I_2/(I_1 + I_2)$ as function of the time delay (note the logarithmic time axis). Every data point corresponds to an average over 20 optical excitations with a non-resonant pump pulse of $\simeq 5$ ns duration. These measurements demonstrate coherences close to 100% and a tuning range of more than $8\pi$ for the phase difference in the internal arms. The lower graph in Fig. 1c shows the total intensity integrated over both output arms as a function of time delay. Similar results as shown in Fig. 1 have previously been obtained in a related polaritonic system[45].

In order to experimentally reveal the underlying physical principles determining the condensation process under non-equilibrium conditions, we will systematically vary the degree of dissipation and feedback in the system. For this purpose, we close one or even both output arms of the interferometer by creating additional potential walls at the ends of the waveguide potentials. Particles that reach these walls are reflected and propagate backwards through the interferometer. This backreflection eventually reaches the location of the Bose–Einstein condensate, where it interferes with the light field present there. Depending on

the phase delays that the particles accumulated on their way, the interference with the Bose–Einstein condensate can be constructive or destructive. Generally speaking, the condensation process is expected to be accelerated in the case of constructive interference, as this increases the local field amplitude, which triggers additional stimulated emission events. In contrast, destructive interference is expected to slow down the growth of the condensate. This feedback phenomenon is fully analogous to effects known from the modified spontaneous emission of quantum emitters[46,47]. The feedback provided by the environment ultimately leads to a situation in which the condensation does not necessarily occur in the lowest energy state as suggested by a gain–loss rule of the Kennard–Stepanov type, but in a condensate state with higher kinetic energy for which the feedback from the environment is more favourable. In the Supplementary Information, we show that the initial evolution of the condensate population upon optical pumping can be approximately described by

$$\frac{\dot{n}}{n} = \Gamma_{em} - \Gamma_{abs}\, e^{\frac{\hbar(\dot{\theta} - \omega_0)}{kT}} - \Gamma_{env}(1 - \mathrm{Re}[r(\dot{\theta})])\ , \qquad (2)$$

where the condensate wavefunction is described by a single complex number $\psi = \sqrt{n}\exp(-i\theta)$ with time-dependent particle number $n = n(t)$ and phase $\theta = \theta(t)$. The interaction with the optical medium is mediated via stimulated absorption and emission with rates $\Gamma_{abs}$ and $\Gamma_{em}$. Furthermore, $\Gamma_{env}$ denotes the rate of photons that are emitted from the condensate to the environment (interferometer), whereas the complex reflection coefficient $r = r(\dot{\theta})$ describes magnitude and phase delay of the coherent feedback created by the closed arm(s) of the interferometer. In general, the reflection coefficient depends on the frequency of the condensate $\dot{\theta}$ as the propagation of particles through the interferometer is highly frequency selective.

Closing the lower output arm (Fig. 2a) reflects the particle flow back through the interferometer and finally to the location of the Bose–Einstein condensate. In Fig. 2b, we show intensity patterns obtained under these conditions for four different time delays after the initial heating pulse. A notable difference to the measurements with the open interferometer is the appearance of standing waves created by the superposition of forward and backward propagating waves. The distance between the nodes in the wavefunction can be taken as a measure of the velocity of the particles. The varying node distances in the upper interferometer arm, e.g., see panel (i) in Fig. 2b, shows that the particles are first accelerated and then decelerated again when they run through the thermo-optically induced potential. In this way, they obtain a different phase delay than the particles in the lower arm.

For the open interferometer, the reflection amplitude $r$ vanishes and the condensate growth rate $g = \dot{n}/n$ in Eq. (2) reduces to the gain–loss rule originating from the Kennard–Stepanov law. In this case, the condensation occurs in the state that minimizes the energy independent of the interferometer configuration. What is effectively found in this situation is an interferometer with an incoming wave having a given, i.e., fixed, frequency, which is the usual case discussed in textbooks. This means that the switching function of the interferometer necessarily needs to assume a sinusoidal shape, as was indeed shown in Fig. 1c. Conversely, any deviation from a sinusoidal switching function indicates that the condensate energy is not a constant during the scanning of the interferometer and, consequently, that the condensation process is not solely determined by an energy minimization rule. In the upper graph of Fig. 2c, we show the normalized intensity in the lower (closed) output arm $\hat{I}_1 = I_1/(I_1 + 2I_2)$ as a function of the time delay in the case of a semi-open interferometer. As we compare the intensity of a travelling wave with that of a standing wave, we have included an additional factor of 2 in the

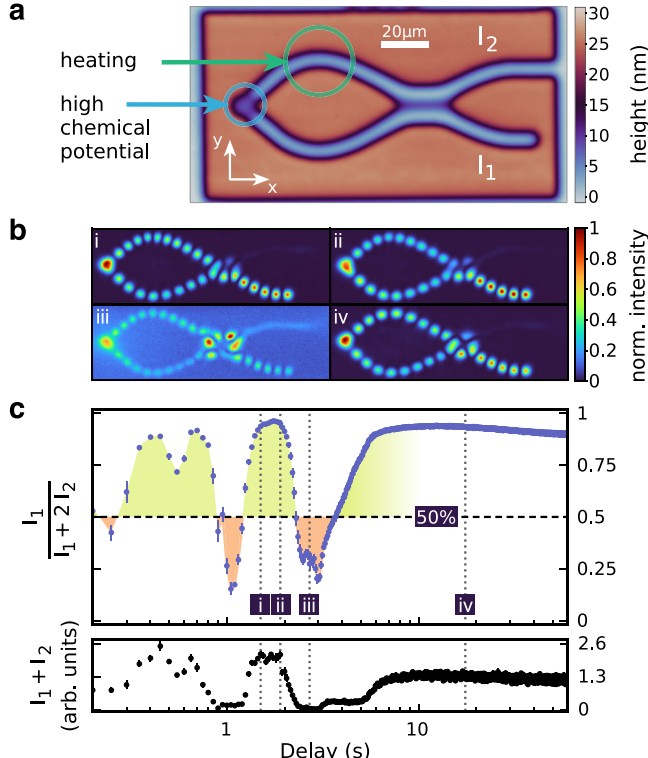

**Fig. 2 Semi-open Mach–Zehnder interferometer. a** Height map of the nanostructured mirror. **b** Normalized photon density for four specific time delays between the heating pulse in the upper internal interferometer arm ('heating') and the optical pumping ('high chemical potential'). The observed standing wave mode patterns indicate a superposition of an outgoing and a back reflected wave. **c** Normalized switching function $I_1/(I_1 + 2I_2)$ (upper graph) and total intensity $I_1 + I_2$ (lower graph) of the semi-open interferometer. Data points are averages over 20 measurements (optical pulses). Error bars indicate the SEM.

normalization factor for compensating the additional intensity contribution of the back-propagating wave in the lower output arm. Interestingly, the switching function in Fig. 2c shows a clear imbalance between the intensities in the upper and lower output arms in favour of the intensity in the lower (closed) output. The latter refers not only to the maximum signal levels in both arms but also to the time intervals for which $\hat{I}_1 > 0.5$. Even if one ignores the data for time delays of $t > 5$ s, as the refractive index of the medium is then largely relaxed, it is observed that the areas shaded in green ($\hat{I}_1 > 0.5$) occupy an area that is more than four times the size of the areas shaded in orange ($\hat{I}_1 < 0.5$). This imbalance shows, in particular, that the condensate energy is not constant, while the interferometer is being scanned. Indeed, for the semi-open interferometer, the condensate growth rate $g = \dot{n}/n$ in Eq. (2) can be shown to favour condensate frequencies that increase the level of constructive interference in the closed arm (see Supplementary Information). As long as it does not become too detrimental in terms of energy (and the correspondingly stronger absorption by the optical medium), the condensate can adjust its kinetic energy in such a way that the particle loss via the open output arm is reduced. The same picture emerges from the behaviour of the total intensity $I_1 + I_2$, which is given as a function of time delay in the lower graph of Fig. 2c. The data indicate that the total intensity is significantly higher when the conditions for constructive interference in the closed arm are met (and vice versa). This simply means that the condensation process experiences a higher growth rate when the particle loss is reduced.

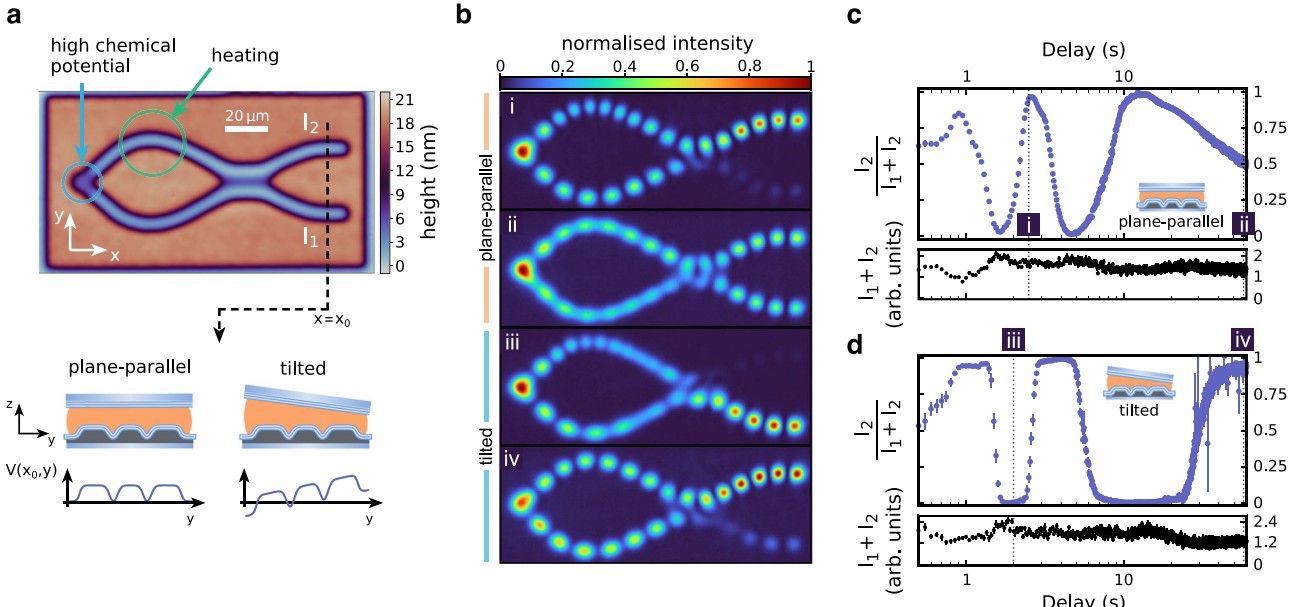

**Fig. 3 Closed Mach–Zehnder interferometer. a** Height map of the nanostructured mirror (top). For the results shown, the microcavity is operated in two configurations: plane-parallel and tilted. By tilting one of the microcavity mirrors, a potential gradient is introduced to the photon gas (bottom). This gradient leads to different phase delays for photons that propagate in the different output arms. **b** Normalized photon density at different time delays for both a plane-parallel and a tilted microresonator. **c** Normalized switching function $I_2/(I_1 + I_2)$ (upper graph) and total intensity $I_1 + I_2$ (lower graph) of the closed interferometer in plane-parallel configuration. **d** Normalized switching function $I_2/(I_1 + I_2)$ (upper graph) and total intensity $I_1 + I_2$ (lower graph) of the closed interferometer in a tilted configuration. Data points are averages over 20 measurements (optical pulses). Error bars indicate the SEM.

For an interferometer that is closed at both output arms (see Fig. 3a), the particles emitted by the condensate return to the location of the condensation after passing through the interferometer twice (see the standing wave density profiles in Fig. 3b). These particles interfere with the condensate, which alters the rate of the condensation process. In the case where the optical path lengths in the two output arms are identical, the system has no reason to prefer constructive or destructive interference in either of the two output arms. Under these conditions, the switching function can be expected to return to the sinusoidal shape as in the case of the open interferometer, which can be confirmed experimentally (see Fig. 3c).

In the case that the optical path lengths in the output arms differ, a different scenario results. Differences in the optical path length of the outputs can be created by tilting one of the cavity mirrors. This effectively adds a potential gradient to the photon gas, which lifts the two output arms to two different potentials (see Fig. 3a). Due to the resulting optical path length difference, the feedback from the two output arms will not interfere in the same way with the condensate. This means, in particular, that the feedback cannot be maximally constructive if the photon density is distributed over both interferometer outputs. Figure 3d shows experimental results for the case of a tilted cavity. The data presented in Fig. 3d reveal an almost discrete switching behaviour between maximum intensity in the upper and maximum intensity in the lower output arm as the phase difference between the internal arms of the interferometer is scanned. This demonstrates that the condensate adjusts its frequency in such a way that a maximum degree of constructive feedback is achieved for all optical path length differences.

Another interesting aspect related to the observed behaviour is the fact that the switching function becomes quite steep, as it jumps between completely destructive and constructive interference. In these regions, the closed interferometer is highly susceptible to changes in optical path length, which is potentially interesting for sensing applications. Indeed, our measurement can

be regarded as a special form of self-mixing interferometry[48–50]. The observation that the switching functions for the plane-parallel and the tilted resonator are different furthermore suggests that the interferometer can also be switched by changing the optical path length in the outputs. This is indeed confirmed experimentally (see Fig. 4). The fact that the frequency of the condensate (and thus the relative intensities in the outputs) can be controlled by a refractive index change that is more than 100 μm away from the location of the condensation clearly demonstrates the non-local character of the condensation process.

In conclusion, our work investigates the Bose–Einstein condensation of photons in a Mach–Zehnder interferometer potential with controlled dissipation and feedback. The switching behaviour of the interferometer is analysed in order to reveal the physical mechanisms that control the formation of a Bose–Einstein condensate. We show that by adjusting their frequency, Bose–Einstein condensates naturally seek to minimize particle loss and destructive interference in their environment. This ability remains hidden in thermal equilibrium, but becomes visible when the condensation occurs under non-equilibrium conditions. Beyond a deeper understanding of Bose–Einstein condensation, our results will be useful for the experimental realization of unconventional computing schemes for the solution of hard optimization problems based on coherent networks of photonic or polaritonic condensates[2,3] and lasers[4,5]. Understanding the physical mechanisms that determine the state of a condensate under controlled dissipation and feedback, as identified in our work, is essential to the design of such systems.

## Methods

**Experimental methods**. Our experiment is based on a high-finesse dye microcavity as shown in Fig. 1a. The mirror separation of $D_0 \simeq 10\ \mu m$ is larger than in earlier photon Bose-Einstein condensate experiments. In general, short mirror spacings create experimental conditions in which the free spectral range of the resonator becomes larger than the emission bandwidth of the dye molecules. The longitudinal mode number then becomes a conserved quantity in the interaction

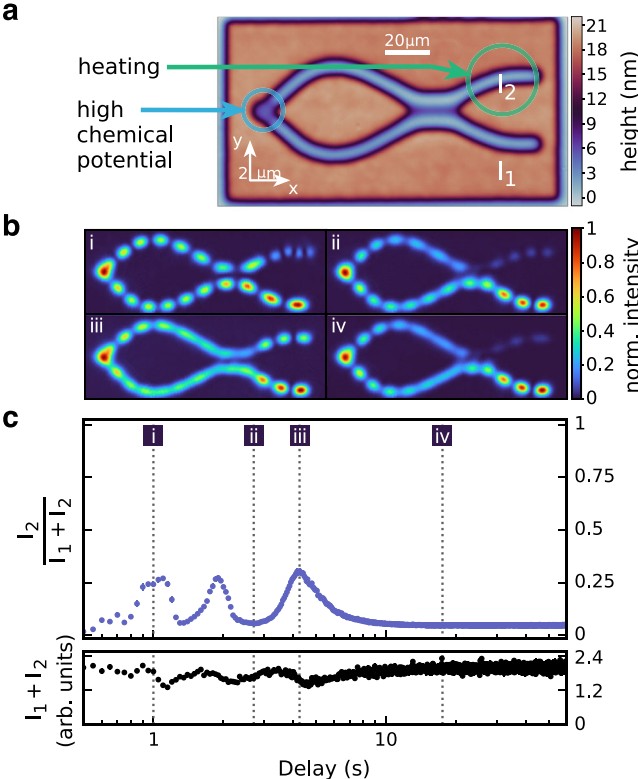

**Fig. 4 Closed Mach–Zehnder interferometer with optical path length tuning in the upper output arm. a** Height map of the nanostructured mirror. **b** Normalized photon density for four specific time delays between the heating pulse in the upper output arm ('heating') and the optical pumping ('high chemical potential'). **c** Normalized switching function $I_2/(I_1 + I_2)$ (upper graph) and total intensity $I_1 + I_2$ (lower graph) of the closed interferometer with output arm tuning. Data points are averages over 20 measurements (optical pulses). Errors (SEM) are smaller than the symbol size.

between the photons and the optical medium. In our present experiments, we find that this is largely the case even at mirror distances where the free spectral range is comparable but not necessarily larger than the dye bandwidth. The advantage of working with a larger mirror spacing results from the reduced pump power required to trigger the condensation process and a lower susceptibility to imperfections in the nanostructuring of the mirror.

In the regime, in which the longitudinal mode number is frozen out, the photon gas becomes effectively 2D. The photon energy in the cavity is given by

$$E = \frac{\hbar c}{n} \sqrt{k_z^2 + k_r^2},\tag{3}$$

in which $n = n(x, y)$ describes the index of refraction and the longitudinal and transverse wavenumbers are denoted by $k_z$ and $k_r$, respectively. The boundary conditions induced by the mirrors require $k_z = \pi q/D$, in which $q$ is the longitudinal mode number and $D = D(x, y)$ denotes the mirror separation across the transverse plane. We assume that both $D(x, y)$ and $n(x, y)$ show only small variations across the cavity plane, i.e., $D(x, y) = D_0 + \Delta d(x, y)$ with $\Delta d(x, y) \ll D_0$ and $n(x,y) = n_0 + \Delta n(x, y)$ with $\Delta n(x, y) \ll n_0$. In a paraxial approximation ($k_r \ll k_z$), the photon energy in Eq. (3) can be rewritten as

$$E \simeq \frac{mc^2}{n_0^2} + \frac{(\hbar k_r)^2}{2m} - \frac{mc^2}{n_0^2}\left(\frac{\Delta d}{D_0} + \frac{\Delta n}{n_0}\right),\tag{4}$$

where $m = \pi\hbar n_0 q/cD_0$ is the effective photon mass. The first term corresponds to the rest energy of the (2D) photon, which is $mc^2/n_0^2 \simeq 2.1$ eV in our experiment (yellow spectral regime). The second term describes the (transverse) kinetic energy and the last term is the potential energy of the photons. The potential energy becomes non-zero, if either the mirror spacing or the refractive index changes locally. In our experiment, variations of the mirror distance can be created by nanostructuring one of the mirrors with a direct laser writing technique. With this technique, a laser scans over the backside of the mirror, locally heating an amorphous silicon layer located below the dielectric stack of the mirror. This causes an irreversible expansion of the silicon layer, pushing the dielectric stack outwards.

In this way, we can create smooth surface structures of up to 100 nm height with sub-nm precision in the growth direction[41]. The height profiles of our mirrors are determined via Mirau interferometry. For these measurements, we use a commercially available interferometric microscope objective (20X Nikon CF IC Epi Plan DI).

The optical medium consists of a solution of rhodamine 6G in water (concentration 10 mmol/L) with a 4% mass fraction of the thermo-responsive polymer pNIPAM added. In our experiments, the dye molecules are excited by a pulsed optical parametric oscillator with a pulse duration of $\simeq$5 ns at a wavelength of 470 nm. This wavelength is outside the reflection band of the cavity mirrors and therefore allows non-resonant excitation on the optical axis. A second pulsed laser at a wavelength of 532 nm with pulse duration 10–20 ms is used to heat the silicon layer on the mirror. This leads to a temperature increase of a few Kelvin, which is enough to reach the LCST of pNIPAM in water. This causes the polymer chains to collapse, which triggers a mass transport in the optical medium that locally changes the refractive index ($\Delta n \simeq 0.1$). In this way, we can create variable potentials for the photon gas[40].

## Data availability
All experimental data used in this study are available in the 4TU.ResearchData database under accession code https://doi.org/10.4121/16566513.v1.

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

## Acknowledgements

This work has received funding from the European Research Council (ERC) under the European Union's Horizon 2020 research and innovation programme (Grant agreement number 101001512).

## Author contributions

M.V. and C.T. performed the experimental studies. All authors carried out the analysis. J.K. and C.T. performed the theoretical modelling. J.K. supervised the work.

## Competing interests

The authors declare no competing interests.
