## [Peer Review File · Nature Communications]

Modified Bose-Einstein condensation in an optical quantum gasREVIEWER COMMENTS

Reviewer #1 (Remarks to the Author):

This is a very nice piece of work, which discovers the non-equilibrium Bose-Einstein condensates of photons in photonic microcavities coupled to Mach-Zehnder interferometer. The nonequilibrium character is revealed by the fact that the condensate wavefunction forms with specific phase due to dynamical equilibrium between gain, loss and the potential landscape. The BEC is formed in such a way that the ratio gain to loss is minimised. I have read the manuscript carefully and I have no comments to the technical part. The paper is well written.

However, my recommendation would be to rewrite the introduction. In particular, the nonequilibrium BECs have been investigated quite a lot in the polariton system. At the end there is not much difference between the polariton BEC and the photon BEC, apart from the fact that relaxation in polariton BEC takes place via interparticle (polariton -exciton, polariton electron) scattering events, whereas in photon BEC it is the process of absorption, relaxation of electronic excitations in the embedded active media followed by reemission. The authors do refer to the polariton work, but briefly at the end of the manuscript, whereas a proper paragraph/description of the history of nonequilibrium polariton BECs should be given in the introduction.

In particular, the nonequilibrium polariton BEC and the excitation spectrum was first described in "Excitations in a Nonequilibrium Bose-Einstein Condensate of Exciton Polaritons

Michiel Wouters and Iacopo Carusotto Phys. Rev. Lett. 99, 140402 – Published 3 October 2007"

The work "Coexisting nonequilibrium condensates with long-range spatial coherence in semiconductor microcavities

Phys. Rev. B 80, 045317 – Published 21 July 2009" reveals for the first time the nonequilibrium nature, by reporting multiple BECs with specific pattern in k-space arising from the dynamical equilibrium between gain and loss, which defines the specific k-space distribution.

The work "Realizing the classical XY Hamiltonian in polariton simulators, Nat. Mater. 16, 1120 (2017) " shows how the nonequilibrium condensate wavefunction is chosen to maximise gain/loss ratio, again the main signature of the non-equilibrium system.

Finally, the non-equilibrium BECs were also described in terms of "weak lasing": "Radiative coupling and weak lasing of exciton-polariton condensates Phys. Rev. B 85, 121301(R) – Published 8 March 2012".

Overall, the polariton system offers great advantage to study condensates in patterned potential landscapes (lattices and circuits), which was extensively investigated experimentally. See "Exciton-polaritons in lattices: A non-linear photonic simulator" C. R.Physique17(2016)934–945

Reviewer #2 (Remarks to the Author):

The manuscript by Vretenar et al. reports on an experimental and theoretical study of a photon condensate that is coupled to an interferometer. They investigate the condensation in several regimes and find that the condensate frequency adjusts itself in order obtain constructive interference between the photon condensate and the light that is fed back into the condensate from the interferometer arms.

The authors claim that this work sheds light on properties of Bose-Einstein condensates, that are hidden in equilibrium.

The paper is well written and the claims seem justified based on the reported experimental results.

I therefore think that the manuscript is suitable for publication in Nature Communications.

the authors may wish to take into account these comments when resubmitting the manuscript.

In the discussion of Fig. 1c, a sinusoidal dependence is claimed. Why is there a sinusoidal dependence on the delay time? Is the phase shift a linear function of the delay time?

- The photons that propagate in the interferometer can also be absorbed and reemitted by the dye molecules, so why is there no energy relaxation toward lower energies? Is the probability of absorption during a roundtrip much smaller than one?

- The claim in the second column on the first page that “This situation, however, still has to be differentiated from laser-like operation in which particle losses overcome photon reabsorption by the optical medium.” is not corroborated by references. Is it really the case that re-absorption in lasers is always small?

- p.2 first paragraph: “If the condensate is created at an elevated potential energy level, for example, particles will gain kinetic energy as they fall into regions of lower potential.”

This has been clearly observed in polariton condensates: see e.g. [E. Wertz, et al. Nat. Phys. 6, 860 (2010)] and theoretically explained [M. Wouters, et al., Phys. Rev. B 77, 115340 (2008)]. In the recent review [J. Bloch et al. arXiv:2106.11137], this effect is put in the perspective of the nonequilibrium nature of polariton and photon condensates.

- the setup is reminiscent of lasers with feedback, such as e.g. in [R. Lang and K. Kobayashi, IEEE JOURNAL OF QUANTUM ELECTRONICS, VOL. QE-16,NO. 3,MARCH 1980]

Reviewer #3 (Remarks to the Author):

In the manuscript titled, “Modified Bose-Einstein condensation in an optical quantum gas” by Vretenar, Toebe and Klaers, the Authors study the Bose-Einstein condensation of photons in an engineered potential landscape that they claim behaves like an effective Mach-Zehnder interferometer. In their experiments, the Authors engineer the potential by nanostructuring the mirror surfaces of the cavity using a direct laser writing technique. By using nonuniform pumping to create the initial condensate, the Authors are able to then control the nonequilibrium condensate and the flow of photons in the potential interferometer by opening and closing the output arms of the interferometer. Optical path differences between the arms of the interferometer are created by either locally heating one of the input arms or by tilting one of the mirrors to create a potential gradient between the two arms.

From my understanding, the key take home message, as framed by the Authors, is that by playing around with the arms of the interferometer and the optical path, the Authors are able to manipulate the flow of photons inside the potential leading to interesting interference effects that reveal the nonequilibrium properties of the photon condensate. To be more detailed, there are three main setups: open, semi-open and closed interferometer, where either none, one or both output arms closed, respectively. For all scenarios, the optical path length in the input is changed by reversibly heating and thus changing the refractive index of one of the arms. This creates the interference expected in a Mach-Zehnder interferometer and depending on the delay between the incoherent pump needed to create the condensate and the pulse to heat the arm, the intensity at the two arms and the interference pattern will vary. For a semi-open or closed system, there is an additional event, the reflection of the photons at the edge of the closed arm. This then leads to interference of reflected photons with the original condensate. The critical result here is that this interaction with the reflected photons changes the loss-gain balance arising from the Kennard-Stepanov relation and forces the nonequilibrium photon gas to condense at a slightly different frequency, thus altering its properties.

I believe this is a fascinating experiment and the key experimental findings here can motivate fellow researchers working on photon BEC to look for other interesting and exotic results, including coherent phenomena, sensing and quantum simulations (as suggested by the Authors). However, one point that lingered in my mind was the broader context of the present version of the work. Although the abstract is framed to sound general, the central presentation of the work appears specialised. Moreover, some of the more general sounding statements/ arguments could do with little more explaining. As such, I believe the Authors should probably try to bring out a more broader picture emanating from the work, which will be of interest to the general Reader, beyond focussing strongly on understanding nonequilibrium condensates, which for me is a very important but a niche topic.

I have some specific comments:

1) The Authors mention in page 2 that the “waveguides are at a lower potential energy.” I am a bit confused by this statement. I would have expected that the photons in the initial condensate to experience the same potential energy in the absence of any horizontal potential bias. So, the statement that they gain kinetic energy as they move along the waveguide is not clear to me. I would request the Authors to clarify this point.

2) In page 4, the Authors state that the fact that condensation occurs in the state that minimises the energy means that the switching function of the interferometer necessarily needs to assume a sinusoidal shape. Why is that the case? I believe the sinusoidal shape reflects the balance of the interferometer as opposed to the semi-open or tilted-closed case. It is later mentioned that the imbalance shows that the condensate energy is not constant. The connection of the sine shaped output or balance with the condensate energy is an important point and should perhaps be explained in more detail.

3) Also, in page 4, I am not sure why the interferometer is an environment here. The total system appears to be closed, and a joint description of the photons in the interferometer and the condensate would be a more apt description, with the dye molecule, polymer and solvent being the external system.

4) In page 5, the statement, “The observation of shoulders in the otherwise sinusoidal shape of the switching function is related to transitions between wave functions with different numbers of nodes,” is not clear to me in the present context.

5) As far as I understand, the mirror is “tilted” to create optical path difference between the output arms in the closed case, in tandem with the optical heating of one of the input arms. In the tilted case, the input arm is subjected to both the mechanisms affecting change in optical length, heating as well as tilting. Does this lead to any observable effect?

On a related note, a more detailed discussion on the effect of longer time delay ($t > 10$), where the behaviour is substantially different would be helpful. Ideally, a longer time delay would entail behaviour where the effect of local heating would be insignificant, as the relevant dynamics has already taken place. But this does not seem to be the case.

6) An important point that I failed to grasp but I believe is central to the main thesis of the paper, is the change in frequency of the condensate when the interferometer arms are manipulated. Is this change observed experimentally? I understand this can be shown using the theoretical model but I still feel a compelling physical argument to explain changes in the loss-gain balance is missing.

7) A minor point. In page 1, the Authors discuss non-equilibrium transport processes in 2D optical gases where condensates propagate in the transverse plane of the resonator by condensing into higher energy states. A similar result was shown in a theoretical work on transport of light in optical microcavities. For an 1D lattice potential structured into one of the mirrors, photons in the cavity transported in the lattice via successive condensation on nearest neighbour lattice sites. Please see, Phys. Rev. A 102, 053517 (2020).

8) Another minor point, in page 2 it is mentioned that the photons recombine after moving apart through the waveguides in a region that is designed as a 50:50 beam-splitter. How is this achieved? Is it simply a result of structuring a meeting point in the mirror?

Reply to Reviewers

We would like to thank the Reviewers for supporting this review process. We are fully aware that the review of a manuscript takes a considerable amount of time that is hardly available in everyday work. With regard to the content of the reviews, we are delighted to see that all three Reviewers are very positive about our work. In the following we will go through the individual points of the reviews. Information describing the changes made in the manuscript are marked in red. The corresponding changes in the manuscript are also in red text colour.

Reviewer #1

1. This is a very nice piece of work, which discovers the non-equilibrium Bose-Einstein condensates of photons in photonic microcavities coupled to Mach-Zehnder interferometer.

We are pleased about this assessment.

2. However, my recommendation would be to rewrite the introduction. In particular, the nonequilibrium BECs have been investigated quite a lot in the polariton system. At the end there is not much difference between the polariton BEC and the photon BEC, apart from the fact that relaxation in polariton BEC takes place via interparticle (polariton -exciton, polariton electron) scattering events, whereas in photon BEC it is the process of absorption, relaxation of electronic excitations in the embedded active media followed by reemission. The authors do refer to the polariton work, but briefly at the end of the manuscript, whereas a proper paragraph/description of the history of nonequilibrium polariton BECs should be given in the introduction.

The reviewer's advice is indeed correct. In our manuscript, we argue that the conclusions drawn from our experiments apply, in principle, to any form of Bose-Einstein condensate. To some extent, this reasoning must also apply the other way round: earlier results on non-equilibrium condensation in other systems are indeed relevant for the present work and should be given more weight in our manuscript. **We have revised the introduction to the manuscript and added the works cited by the reviewer to our list of references. Please see also our answer to point 2 of Reviewer #3, which also deals with the introduction.**

Reviewer #2

1. I therefore think that the manuscript is suitable for publication in Nature Communications.

We thank the Reviewer for his or her recommendation.

2. In the discussion of Fig. 1c, a sinusoidal dependence is claimed. Why is there a sinusoidal dependence on the delay time? Is the phase shift a linear function of the delay time?

Figure 1c serves several purposes. On the one hand, it should be demonstrated that the potential landscape that we create through mirror nanostructuring actually works as a Mach-Zehnder interferometer. Furthermore, we obtain information about how the phase delay behaves as a function of time. When interpreting Fig. 1c, one must keep in mind that the time axis is plotted logarithmically. The approximately sinusoidal switching function of the interferometer means that the phase shift changes linearly with logarithmic time, or equivalently, exponentially with real time. **In order to draw the reader's attention to the logarithmic time axis, we now explicitly refer to it both in the text and in the figure caption.** The origins of this exponential decay lie in the physics of the thermosensitive polymer used and are not entirely obvious. Generally speaking, however, an exponential decay is not uncommon in a relaxation process.

3. The photons that propagate in the interferometer can also be absorbed and reemitted by the dye molecules, so why is there no energy relaxation toward lower energies? Is the probability of absorption during a roundtrip much smaller than one?

The rate of photon reabsorption in the interferometer is the same as at the point of condensation, since neither the properties of the dye medium nor the frequency of the light field vary across the system. The fact that the condensation does not take place at the location of the lowest energy, which is located in the arms of the interferometer, is due to an interplay of the finite mirror finesse and spatially inhomogeneous optical pumping.

Thermal equilibrium in the photon-dye system means that the photon energies are Bose-Einstein distributed and the optical medium has a spatially homogeneous excitation level, which is equivalent to the statement that the chemical potential of the photons is the same everywhere. If all losses could be eliminated, any initially existing gradients in the photon chemical potential would disappear over time. This relaxation time, however, depends on how inhomogeneous the chemical potential is initially. If the electronic excitations are spatially concentrated close to one point, for example, the relaxation will most likely take a long time. If the system starts from an almost spatially homogeneous excitation level, the relaxation will be fast.

For real systems photon losses come into play, but the general physics remains similar to that just described. In case of spatially homogeneous optical pumping, it is still “easy” for the system to relax into equilibrium [on the timescale given by the reabsorption time, see Schmitt et al., Phys. Rev. A 92, 011602 (2015)]. With spatially inhomogeneous optical pumping, however, the system no longer necessarily comes into global equilibrium due to the losses it experiences during the relaxation. In other words, just by the type of optical pumping one can prepare a system that is either close to thermal equilibrium or that is very far from it [see, e.g., Marelic et al., Phys. Rev. A 91, 033813 (2015)]. In the present work we deliberately examine the case of spatially highly inhomogeneous optical pumping that prevents the system from relaxing into a global thermal equilibrium.

4. The claim in the second column on the first page that “This situation, however, still has to be differentiated from laser-like operation in which particle losses overcome photon reabsorption by the optical medium.” is not corroborated by references. Is it really the case that re-absorption in lasers is always small?

Reabsorption in a laser is not always small (in the sense of negligible). With our statement we aim to highlight the following difference: When designing lasers, one typically tries to prevent reabsorption since it increases the laser threshold. Reabsorption is the reason why 4-level lasers are easier to operate than 3-level lasers. With a 4-level laser design, one can basically switch off the reabsorption of the laser radiation by the optical medium, which allows for lower thresholds. In the case of photon condensation experiments, on the other hand, one tries as best as possible experimentally to prevent light from escaping from the resonator (similar as in a black body radiator). To achieve this, mirrors with a very high reflectivity (up to 99.9985% in our case) and an optical medium with a strong spectral overlap between absorption and emission are used. The approach is to maximize reabsorption, which is the exact opposite of what you typically try to do with a laser. **Based on the Reviewer’s comment, we have rephrased the corresponding sentence so that it should be clearer what the role of reabsorption is for lasing and for condensation experiments.**

5. p.2 first paragraph: “If the condensate is created at an elevated potential energy level, for example, particles will gain kinetic energy as they fall into regions of lower potential.” This has been clearly observed in polariton condensates: see e.g. [E. Wertz, et al. Nat. Phys. 6, 860 (2010)] and theoretically explained [M. Wouters, et al., Phys. Rev. B 77, 115340 (2008)]. In the recent review [J. Bloch et al. arXiv:2106.11137], this effect is put in the perspective of the nonequilibrium nature of polariton and photon condensates.

We suspect that the comment is aimed at the repulsive self-interactions of exciton-polaritons, which lead to an acceleration away from the location of the optical pumping due to a local blue shift of the effective potential landscape. The situation in the photon BEC system is different, however. The instantaneous self-interactions of the photons are negligible for all practical purposes (the only form of self-interactions that play a certain role are thermo-optical interactions on very slow time scales). In our case, the acceleration of the photons into the transverse plane of the resonator only comes about by intentionally creating a corresponding potential step in the height profile of the mirror used. **We took the reviewer's comment as an opportunity to point out this difference between the polariton and the photon BEC system in our manuscript.**

6. The setup is reminiscent of lasers with feedback, such as e.g. in [R. Lang and K. Kobayashi, IEEE JOURNAL OF QUANTUM ELECTRONICS, VOL. QE-16,NO. 3,MARCH 1980]

As stated in our manuscript, our experiment can be seen as a special form of self-mixing interferometry (which is one of the terms that is used to describe certain feedback schemes for lasers). The corresponding references given in our manuscript are M. Rudd, Journal of Physics E: Scientific Instruments 1, 723 (1968) and G. Giuliani et al., Journal of Optics A: Pure and Applied Optics 4, S283 (2002). **The paper mentioned by the Reviewer fits indeed well into this context, which is why we have included it as an additional reference.**

Reviewer #3

1. I believe this is a fascinating experiment and the key experimental findings here can motivate fellow researchers working on photon BEC to look for other interesting and exotic results, including coherent phenomena, sensing and quantum simulations (as suggested by the Authors).

We thank the Reviewer for this very positive assessment of our manuscript.

2. However, one point that lingered in my mind was the broader context of the present version of the work. Although the abstract is framed to sound general, the central presentation of the work appears specialised. Moreover, some of the more general sounding statements/ arguments could do with little more explaining. As such, I believe the Authors should probably try to bring out a more broader picture emanating from the work, which will be of interest to the general Reader, beyond focussing strongly on understanding nonequilibrium condensates, which for me is a very important but a niche topic.

To some extent, we find it inevitable that the presentation becomes narrower in the middle section of the manuscript, which naturally has the task of describing the details of the experiment carried out. This also reflects a conscious decision on our part to separate the description of the experiment and its interpretation and embedding in a larger context (introduction and last paragraph). However, we absolutely agree with the Reviewer that the embedding the experiment in a larger context is of great importance in order to appeal to the largest possible readership.

If we had to sum up our work in one sentence, it would be this: By adjusting their frequency Bose-Einstein condensates naturally try to avoid particle loss and destructive interference in their environment. We consider this to be a fairly general finding about the nature of Bose-Einstein condensation. This finding may not be entirely unexpected to everyone, but we believe it has never been shown in a similarly systematic manner. **In revising our abstract and introductory paragraph, we tried to emphasize this main result, which we think should be of interest to anyone interested in understanding Bose-Einstein condensation. In addition, we provide further information on the current state of the art in this area (see the comment by Reviewer # 1). Also, the connection to current activities in spin glass simulation with optical condensates should become clearer now. But the truth is also that we have not fundamentally changed the introduction, as we consider it suitable to motivate the topic of this work.**

3. The Authors mention in page 2 that the “waveguides are at a lower potential energy.” I am a bit

confused by this statement. I would have expected that the photons in the initial condensate to experience the same potential energy in the absence of any horizontal potential bias. So, the statement that they gain kinetic energy as they move along the waveguide is not clear to me. I would request the Authors to clarify this point.

The photons emitted by the condensate gain kinetic energy only when they enter the waveguide potentials. This is caused by the potential step that separates the condensate from the waveguide potential. When the photons propagate in the waveguide, no additional acceleration takes place (except for the thermo-optical potential in the upper interferometer arm). **We have reworded the corresponding statement to emphasize that the acceleration of photons takes place when they are emitted from the condensate (but not necessarily within the waveguide potentials).**

4. In page 4, the Authors state that the fact that condensation occurs in the state that minimises the energy means that the switching function of the interferometer necessarily needs to assume a sinusoidal shape. Why is that the case? I believe the sinusoidal shape reflects the balance of the interferometer as opposed to the semi-open or tilted-closed case. It is later mentioned that the imbalance shows that the condensate energy is not constant. The connection of the sine shaped output or balance with the condensate energy is an important point and should perhaps be explained in more detail.

For the open interferometer, there is no feedback from the environment that could alter the condensation process. Under such conditions, the only relevant criterion is the (kinetic) energy of the condensate, which the condensation process tries to minimize. What you effectively find in this situation is an interferometer with an incoming wave that has a given (i.e., fixed) frequency. This is the usual case discussed in textbooks. It is clear that a sinusoidal switching function of the interferometer is expected. **We have modified the relevant passage to further clarify our statements.**

5. Also, in page 4, I am not sure why the interferometer is an environment here. The total system appears to be closed, and a joint description of the photons in the interferometer and the condensate would be a more apt description, with the dye molecule, polymer and solvent being the external system.

The distinction between system and environment is primarily a convention. In the case of the open or semi-open interferometer, this convention may seem natural, but perhaps less so for the closed interferometer. Our division into system and environment is mainly motivated by our theoretical analysis of the experiment (see Theoretical Methods) in which we describe the influence of the environment on the condensate using a single complex-valued reflection function. This approach works regardless of whether the interferometer is open or closed and allows us to qualitatively reproduce and explain our experimental findings. In general, however, other divisions between system and environment are certainly possible, as suggested by the Reviewer, depending on what purpose this is to serve.

6. In page 5, the statement, "The observation of shoulders in the otherwise sinusoidal shape of the switching function is related to transitions between wave functions with different numbers of nodes," is not clear to me in the present context.

We agree with the Reviewer that our statement is a bit cryptic. What we intended to do here was to point out that there are deviations from a perfect sinusoidal shape which, if you look more closely at the data, can be traced back to the fact that the closed interferometer shows an effectively discrete mode spectrum. Ultimately, however, this is not a very important point. It seems to us that the best solution is not to highlight this point at all, so as not to divert attention from the main points of this paragraph. We have accordingly removed the sentence.

7. As far as I understand, the mirror is "tilted" to create optical path difference between the output arms in the closed case, in tandem with the optical heating of one of the input arms. In the tilted case, the input arm is subjected to both the mechanisms affecting change in optical length, heating as well as tilting. Does this lead to any observable effect?

As the Reviewer correctly writes, the cavity tilt indeed gives an offset in the pathlength difference of the internal interferometer arms. However, this has no significant effect on our experiments since the phase difference between the internal arms is scanned by the thermo-optical potential anyway.

8. On a related note, a more detailed discussion on the effect of longer time delay ($t > 10$), where the behaviour is substantially different would be helpful. Ideally, a longer time delay would entail behaviour where the effect of local heating would be insignificant, as the relevant dynamics has already taken place. But this does not seem to be the case.

Generally speaking, we observe that the phase difference between the interferometer arm vanishes exponentially with the delay time (see also point 2 in the response to Reviewer #2). From a certain point in time, however, the thermo-optical potential has almost completely disappeared, so that the phase difference becomes almost constant over time (> 10 s). The switching of the interferometer then simply stops. Accordingly, we do not expect any unexpected effects for longer times.

9. An important point that I failed to grasp but I believe is central to the main thesis of the paper, is the change in frequency of the condensate when the interferometer arms are manipulated. Is this change observed experimentally? I understand this can be shown using the theoretical model but I still feel a compelling physical argument to explain changes in the loss-gain balance is missing.

Indeed, that is a central point of our experiment. We do indeed measure the frequency of the condensate, but not with the help of an external dispersive optical element (like a grating spectrometer) but with the help of the intra-cavity Mach-Zehnder interferometer. There is a direct connection between the frequency of the condensate and the splitting ratio of the interferometer outputs (at a given optical path length difference between the input arms). A constant condensate frequency, for example, implies a sinusoidal switching function. An imbalanced switching function can only occur, if the condensate frequency varies as the interferometer is being scanned. Further details are given in the "Theoretical Methods".

10. A minor point. In page 1, the Authors discuss non-equilibrium transport processes in 2D optical gases where condensates propagate in the transverse plane of the resonator by condensing into higher energy states. A similar result was shown in a theoretical work on transport of light in optical microcavities. For an 1D lattice potential structured into one of the mirrors, photons in the cavity transported in the lattice via successive condensation on nearest neighbour lattice sites. Please see, Phys. Rev. A 102, 053517 (2020).

We have included this work to the list of references.

11. Another minor point, in page 2 it is mentioned that the photons recombine after moving apart through the waveguides in a region that is designed as a 50:50 beam-splitter. How is this achieved? Is it simply a result of structuring a meeting point in the mirror?

The 50:50 beam splitter is implemented by a potential barrier between the waveguides in which the potential level is chosen so that the incident waves are transmitted as strongly as they are reflected. The most important parameter here is the height of the potential barrier in the splitter. Preliminary experimental tests were carried out to find the right parameter for a nearly 50:50 splitting ratio. Ultimately, the found potential barrier height is quite comparable with the potential level at the location of the condensate. This is also physically understandable, since the incoming particles then almost come to a standstill at the top of the potential barrier and "fall down" equally on both sides.

REVIEWERS' COMMENTS

Reviewer #1 (Remarks to the Author):

The authors modified the introduction following my recommendations and I am now happy to recommend the paper for publication

Reviewer #2 (Remarks to the Author):

The authors have given very satisfactory responses to all the comments/questions raised by the referees and sill improved the manuscript. I therefore recommend publication in Nat. Comm.

Reviewer #3 (Remarks to the Author):

In the revised manuscript titled, "Modified Bose-Einstein condensation in an optical quantum gas" by Vretenar, Toebes and Klaers, the Authors have clearly responded to the questions raised by the reviewers, and made suitable changes to the main text where necessary.

As mentioned in my previous report, which also seems to be the opinion of my fellow reviewers, I think this is a fascinating experiment that will motivate interested groups and researchers to pursue related work. As such, I recommend acceptance of the paper in its present form.

On a minor note, the version of the draft I downloaded from the Nat. Commun. website has several errors and typos, including missing references. But I am assuming these will be corrected during the final copyediting process.

Reply to Reviewer

Reviewer #3 has noted that the draft *“downloaded from the Nat. Commun. website has several errors and typos, including missing references.”*

We have thoroughly worked through the entire text again. We did not find any obvious spelling errors in the main text itself. However, we did notice that the text in the Figures contained American spelling, while the rest of the manuscript was in British spelling. We have adjusted the spelling in the figures accordingly.

It is not entirely clear to us where references could be missing in our manuscript. We went through all the references again and optimised their order in the text. However, in our opinion there are no errors here. What is indeed missing is the link to the experimental data [stored at 4TU.ResearchData]. This can be provided in the next few days.